# Cytokine ranking via mutual information algorithm correlates cytokine profiles with presenting disease severity in patients infected with SARS-CoV-2

Kelsey E Huntington[1,2,3,4,5,6,7†], Anna D Louie[1,2,3,4,8†], Chun Geun Lee[3,4,7], Jack A Elias[3,4,7], Eric A Ross[9], Wafik S El-Deiry[1,2,3,4,5,6,10]*

[1]Brown Experimentalists Against COVID-19 (BEACON) Group, Brown University, Providence, United States; [2]Laboratory of Translational Oncology and Experimental Cancer Therapeutics, Warren Alpert Medical School, Brown University, Providence, United States; [3]The Joint Program in Cancer Biology, Brown University and Lifespan Health System, Providence, United States; [4]Cancer Center at Brown University, Warren Alpert Medical School, Brown University, Providence, United States; [5]Department of Pathology and Laboratory Medicine, Warren Alpert Medical School, Brown University, Providence, United States; [6]Pathobiology Graduate Program, Warren Alpert Medical School, Brown University, Providence, United States; [7]Department of Molecular Microbiology and Immunology, Brown University, Providence, United States; [8]Department of Surgery, Lifespan Health System and Warren Alpert Medical School, Brown University, Providence, United States; [9]Biostatistics and Bioinformatics Facility, Fox Chase Cancer Center, Temple University Health System, Philadelphia, United States; [10]Hematology-Oncology Division, Department of Medicine, Lifespan Health System and Warren Alpert Medical School, Brown University, Providence, United States

*For correspondence:
wafik@brown.edu

†These authors contributed equally to this work

**Abstract** Although the range of immune responses to severe acute respiratory syndrome coronavirus 2 (SARS-CoV-2) is variable, cytokine storm is observed in a subset of symptomatic individuals. To further understand the disease pathogenesis and, consequently, to develop an additional tool for clinicians to evaluate patients for presumptive intervention, we sought to compare plasma cytokine levels between a range of donor and patient samples grouped by a COVID-19 Severity Score (CSS) based on the need for hospitalization and oxygen requirement. Here we utilize a mutual information algorithm that classifies the information gain for CSS prediction provided by cytokine expression levels and clinical variables. Using this methodology, we found that a small number of clinical and cytokine expression variables are predictive of presenting COVID-19 disease severity, raising questions about the mechanism by which COVID-19 creates severe illness. The variables that were the most predictive of CSS included clinical variables such as age and abnormal chest x-ray as well as cytokines such as macrophage colony-stimulating factor, interferon-inducible protein 10, and interleukin-1 receptor antagonist. Our results suggest that SARS-CoV-2 infection causes a plethora of changes in cytokine profiles and that particularly in severely ill patients, these changes are consistent with the presence of macrophage activation syndrome and could furthermore be used as a biomarker to predict disease severity.

## Introduction

In December 2019, severe acute respiratory syndrome coronavirus 2 (SARS-CoV-2), the origin of coronavirus disease 2019 (COVID-19), emerged in Wuhan, China (*Zhu et al., 2020*). Although many COVID-19 patients remain asymptomatic, there exists a subset of patients who present with severe illness. Early treatment with dexamethasone appears to improve outcomes in these patients. However, it is not always initially clear which patients would benefit from this therapy (*The RECOVERY Collaborative Group, 2020*). Moreover, COVID-19 infection can be accompanied by a severe inflammatory response characterized by the release of pro-inflammatory cytokines, an event known as cytokine storm (CS) (*Tang et al., 2020*; *Ragab et al., 2020*). Thus far, this COVID-19-associated CS has predominantly been characterized by the presence of IL-1β, IL-2, IL-17, IL-8, TNF, CCL2, and most notably IL-6 (*Tang et al., 2020*; *Merad and Martin, 2020*; *McGonagle et al., 2020*; *Wan et al., 2020*; *Otsuka and Seino, 2020*). Severe cases of CS can be life threatening, and early diagnosis as well as treatment of this condition can lead to improved outcome. We hypothesize that cytokine profiles combined with clinical information can predict disease severity, potentially giving clinicians an additional tool when evaluating patients for preemptive intervention.

## Results

Analysis was performed for 36 PCR-confirmed COVID-19 (+) and 36 (−) human plasma samples (*Figure 1—source data 1*). The COVID-19 Severity Score (CSS) was developed to categorize patients based on their status upon presentation to the emergency department. CSS is graded as follows: 0 = COVID (−), no symptoms, healthy control (n = 24); 1 = COVID (−), symptoms (n = 12); 2 = COVID (+), discharged from emergency room (n = 15); 3 = COVID (+), admitted, but who did not require supplemental oxygen (n = 7); 4 = COVID (+), admitted and required any amount of supplemental oxygen or positive pressure ventilation (n = 8); and 5 = COVID (+), admitted to ICU/step-down (n = 6) (*Figure 1*). CSS was used as the outcome variable for a mutual information minimum-redundancy maximum-relevance algorithm (*Kratzer and Furrer, 2018*; *Figure 1*), with the goal of selecting a subset of variables most predictive of CSS. The algorithm confirmed the predictive value of clinical variables such as age and chest x-ray abnormality and also ranked the information gain provided by each of 15 cytokines tested. Several cytokines were able to add unique predictive value to the mutual information model in addition to what was provided by clinical factors such as age or patient comorbidities. This algorithm also deprioritized factors when their predictive value was redundant with the most predictive variables. Macrophage colony-stimulating factor (M-CSF) was ranked second after age as it was the factor that added the most predictive power to the algorithm with minimal redundancy with age. It ranked ahead of abnormalities on chest x-ray because while both were relevant in predicting COVID severity, part of the predictiveness of chest x-ray abnormality was also explained by age differences (*Figure 2*). The top four cytokines combined with age were predictive of the most severe CSS (4–5) and had a receiver operating characteristic (*Figure 2*), with an area under the curve of 0.86. Multiple cytokines, including M-CSF (p<0.01), interferon-inducible protein 10 (IP-10) (p<0.01), interleukin 18 (IL-18) (p<0.01), and interleukin-1 receptor antagonist (IL-1RA) (p<0.01), were more relevant in predicting CSS than more frequently characterized cytokines in the context of COVID-19 such as IL-6 (p<0.01). These cytokines showed a statistically significant difference in their profiles when segregated by CSS (*Figure 3*), yet the mutual information algorithm prioritized them differently than would be expected based on univariate analyses. This indicates that the mutual information algorithm is prioritizing cytokines whose predictive value for COVID-19 severity cannot be fully explained by other clinical variables such as age or medical comorbidities.

## Discussion

We found that a small number of clinical variables when combined with cytokine expression are predictive of presenting COVID-19 disease severity. Cytokines singled out for relevance by the mutual information algorithm shared a connection to macrophage activation syndrome (MAS), raising questions about the mechanism by which SARS-CoV-2 creates severe illness in a subset of patients. First, we examined the significant contribution of IP-10 to CSS. IP-10 is secreted by monocytes, fibroblasts, and endothelial cells in response to interferon gamma (IFN-γ), which is secreted by T cells

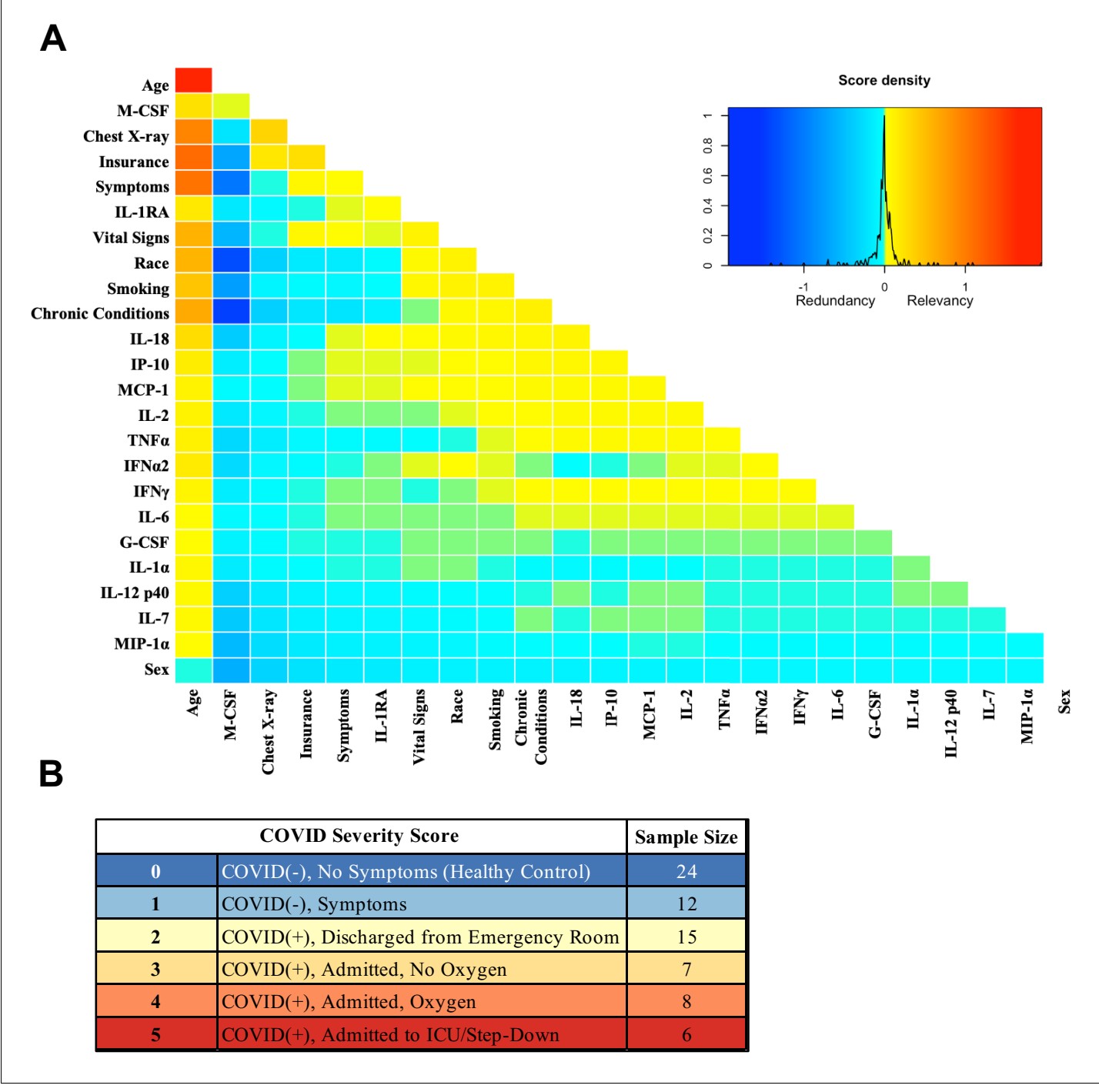

**Figure 1.** Mutual information COVID Severity Score (CSS) relevancy matrix. (**A**) Comprehensive matrix of relevancy to CSS of all variables assessed by mutual information algorithm, relevancy scores computed for not-yet selected variables are shown in each column, and variables are ordered to place maximum local scores on the diagonal, yielding a list in decreasing order from the upper left of variable relevancy. Warmer colors indicate higher relevancy, while cooler colors indicate higher redundancy. (**B**) COVID Severity Score Table with breakdown of categories as well as sample size per category.

The online version of this article includes the following source data for figure 1:

**Source data 1.** Mutual algorithm criteria table.

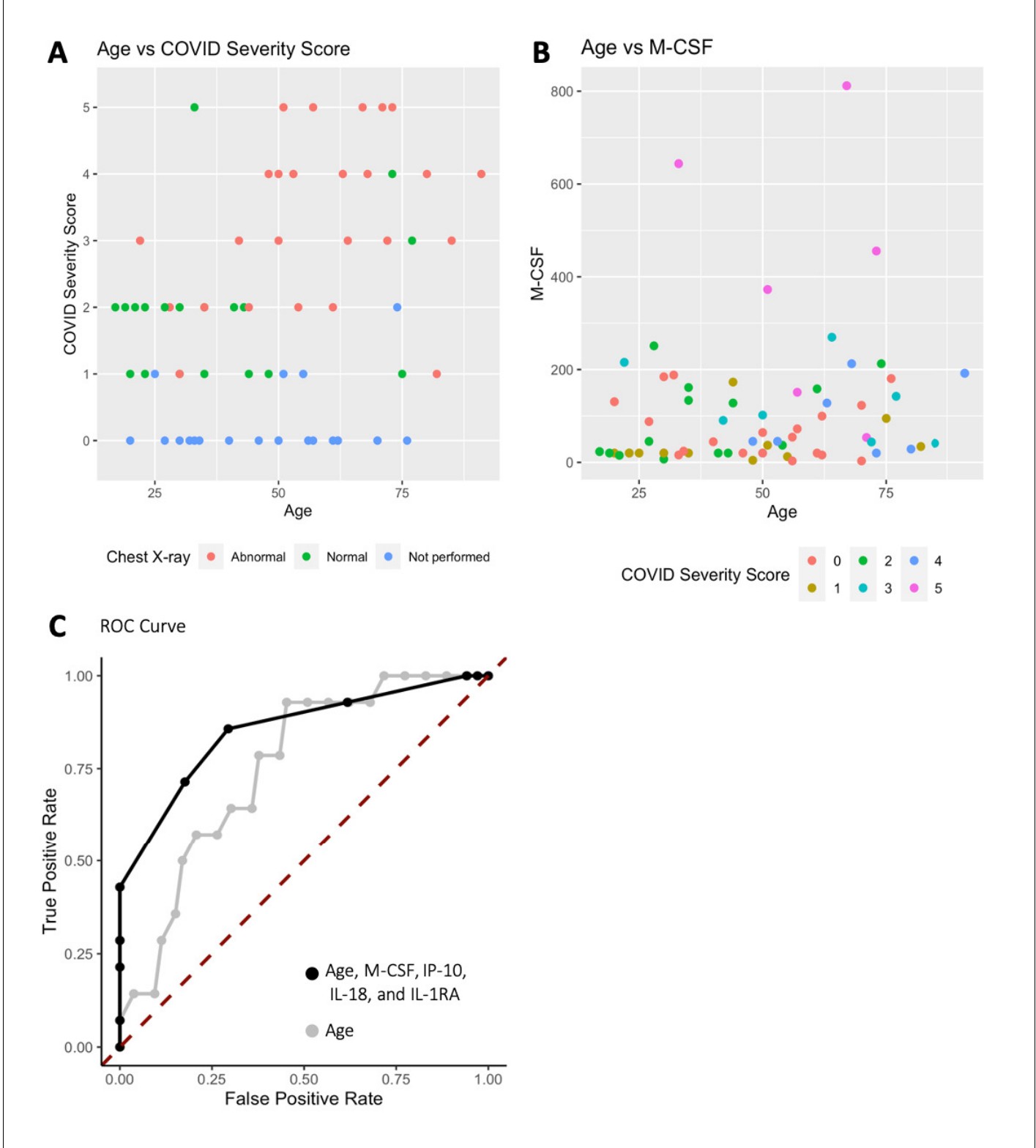

**Figure 2.** Age, macrophage colony-stimulating factor (M-CSF), and chest x-ray are the most predictive variables for COVID Severity Score (CSS). (**A**) The y-axis is CSS, and the x-axis is age in years with points colored by chest x-ray status. (**B**) The y-axis is M-CSF concentration in pg/mL, and the x-axis is age in years, with points colored based on CSS. See individual legends below graphs. (**C**) Receiver operating characteristic (ROC) curve predicting CSS 4–5 using age, M-CSF, IP-10, IL-18, and IL-1RA, and ROC curve predicting CSS 4–5 using only age.

The online version of this article includes the following source data for figure 2:

**Source data 1.** Patient information.

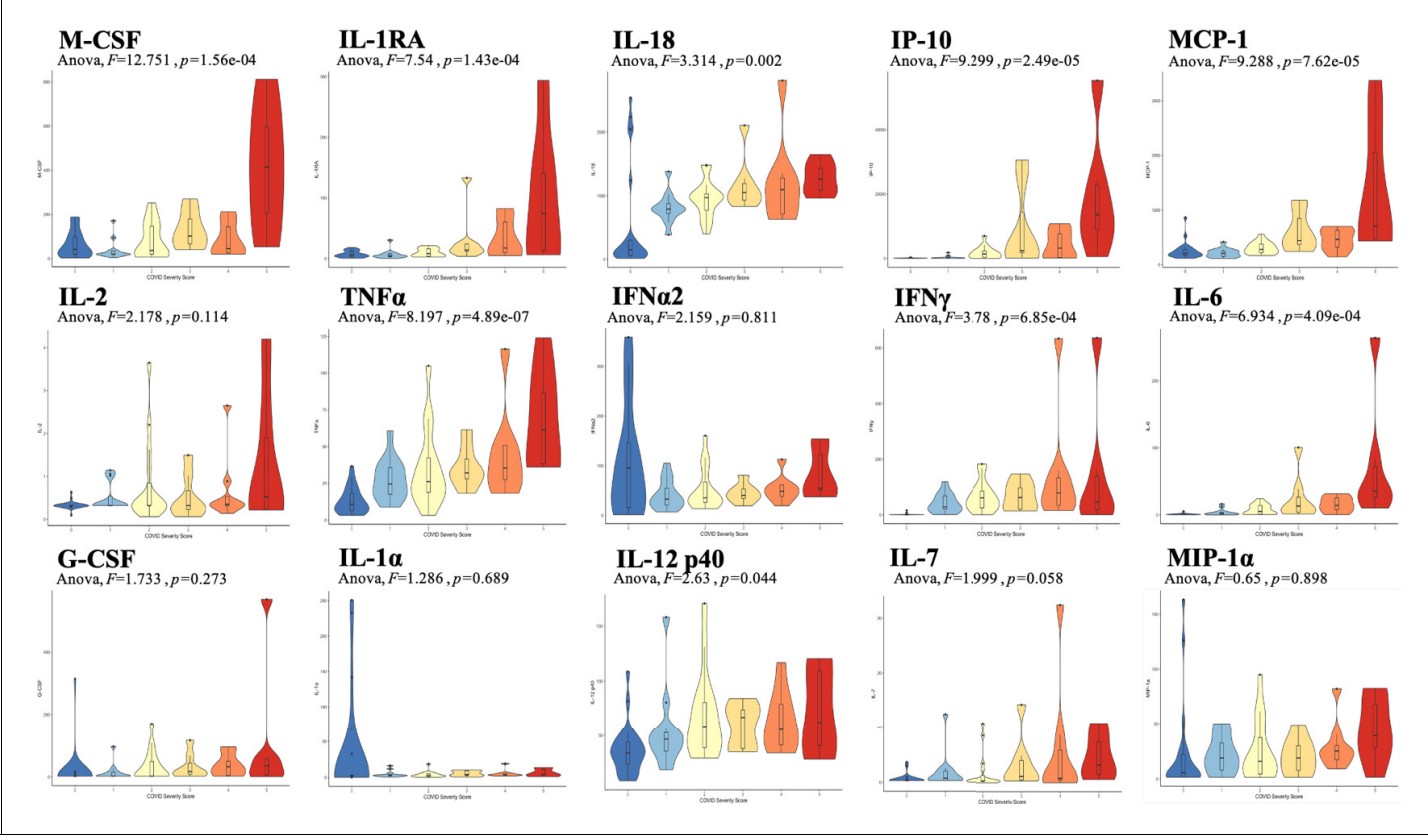

**Figure 3.** Violin plot representations of cytokine expression levels ordered by COVID Severity Score (CSS). Cytokines ordered by row from upper left corner based on mutual information relevancy matrix (upper left being most relevant and lower right being least relevant). The x-axis is CSS, and the y-axis is analyte concentration in pg/mL. One-way ANOVA F values and p values are listed on each plot. *Source code 1*. R code with sections to apply the varrank package to the source data, create a receiver operating characteristic curve, and calculate analysis of variance for each cytokine. The online version of this article includes the following source data for figure 3:

**Source data 1.** Raw data.

(mainly, Th1), macrophages, mucosal epithelial cells, and natural killer (NK) cells (*Liu et al., 2011*). This release of IFN-γ induces several cell types to produce IP-10, which consequently recruits more Th1 cells, contributing to a positive feedback loop. IP-10 is also chemoattractant to CXCR3-postitive cells such as macrophages, dendritic cells, NK cells, and T cells. It has been proposed that macrophages recruited by IP-10, in the presence of persistent IFN-γ production, can lead to MAS (*Merad and Martin, 2020*; *McGonagle et al., 2020*; *Otsuka and Seino, 2020*). MAS is characterized as a state of systemic hyperinflammation often accompanied by CS, which, without intervention, can lead to severe tissue damage and, in extreme cases, death (*Otsuka and Seino, 2020*).

Moreover, the cytokine most relevant in predicting CSS was M-CSF, which is secreted by eukaryotic cells in response to viral infection and stimulates hematopoietic stem cells to differentiate into macrophages. Currently, there are three separate immune stages that describe the progression of COVID-19. The first stage is characterized by a potent induction of interferons that marks the early activation of the immune system that is important in the viral response, and the second stage is characterized by a delayed interferon response (*Merad and Martin, 2020*). These stages may prime the body for a third stage comprised of detrimental hyperinflammation characterized by CS and MAS (*Merad and Martin, 2020*). This excessive macrophage activation could explain the increase in IL1-RA that we observed, a cytokine abundantly produced by macrophages.

Steroids have shown a survival benefit for COVID-19, likely by suppressing such detrimental hyperinflammation (*The RECOVERY Collaborative Group, 2020*). Our analysis identified a pattern of cytokine alterations on presentation associated with COVID-19 severity. The ability to identify a cytokine pattern less redundant with known clinical factors such as age and chest x-ray could help

better identify patients in need of immunomodulatory treatment without the confounders of current models where the measured cytokines correlate as much with age as with severity (*Pierce et al., 2020*). Further studies should be conducted to clarify the mechanistic role that these cytokines and macrophages play in the various stages of COVID-19 and correlate them with other hematologic parameters that were not collected in this database. The results of these future studies could identify more targeted immunomodulatory strategies beyond steroid administration such as treatment with MEK inhibitors (*Zhou et al., 2020*), as well as the ideal timing of these interventions to maximize therapeutic efficacy. Future studies could also address the size limitations of this study, which was not powered to explore race- or ethnicity-related differences in COVID-19 severity. Finally, we present the application of this mutual information algorithm as a way to evaluate the dataset as a whole and elucidate the most important cytokines in predicting the presenting severity of COVID-19. COVID-19 severity is influenced by many clinical factors, such as age, and this algorithm is able to identify cytokines that contribute information not present in the tested clinical variables. Identifying the most important variables for severe presentation of COVID-19 within a more complete cytokine profile may help determine global immune mechanisms of disease severity.

# Materials and methods

## Key resources table

| Reagent type (species) or resource | Designation | Source or reference | Identifiers | Additional information |
|---|---|---|---|---|
| Biological samples (*Human*) | Human plasma | Lifespan Brown COVID-19 Biobank | See data table | 48 unique patients |
| Biological samples (*Human*) | Human plasma | Lee BioSolutions | 991–58-PS | 24 unique patients |
| Commercial assay or kit | MILLIPLEX MAP Human Immunology Multiplex Assay | Millipore Sigma | 15-Plex # HCYTA-60K | |
| Software, algorithm | R | R Project for Statistical Computing | RRID:SCR_001905 | |
| Other | Luminex 100/200 System assay platform | Thermo Fisher Scientific | https://www.luminexcorp.com/luminex-100200 | |

## Biobank samples

COVID-19 (+) and (−) human plasma samples were received from the Lifespan Brown COVID-19 Biobank from Brown University at Rhode Island Hospital (Providence, RI). All biobank samples were collected on patients' arrival in the Emergency Department at Rhode Island Hospital. All patient samples were deidentified but included the available clinical information as described in Results. It is unknown if any patients were blood relatives. The IRB study protocol 'Pilot Study Evaluating Cytokine Profiles in COVID-19 Patient Samples' did not meet the definition of human subjects research by either the Brown University or the Rhode Island Hospital IRBs. All samples were thawed and centrifuged at 14,000 rpm for 10 min following the manufacturer protocol included with the Luminex kit to remove cellular debris immediately before the assay was run.

## Donor samples

Normal, healthy, COVID-19 (−) samples were commercially available from Lee BioSolutions (991–58-PS-1, Lee BioSolutions, Maryland Heights, MO). All samples were thawed and centrifuged at 14,000 rpm for 10 min following the manufacturer protocol included with the Luminex kit to remove cellular debris immediately before the assay was run.

## Cytokine and chemokine measurements

A MilliPlex MILLIPLEX MAP Human Cytokine/Chemokine/Growth Factor Panel A – Immunology Multiplex Assay (HCYTA-60K-13, Millipore Sigma, Burlington, MA) was run on a Luminex 200 Instrument (LX200-XPON-RUO, Luminex Corporation, Austin, TX) according to the manufacturer's instructions. Plasma levels of granulocyte colony-stimulating factor (G-CSF), IFN-γ, interleukin one alpha (IL-1α),

interleukin-1 receptor antagonist (IL-1RA), IL-2, IL-6, IL-7, IL-12, IP-10, monocyte chemoattractant protein-1 (MCP-1), M-CSF, macrophage inflammatory protein-1 alpha (MIP-1α), and tumor necrosis factor alpha (TNF-α) were measured. Data pre-processing: values below limit of detection were re-coded as half the limit of detection. A single extreme outlier value in IFN-y levels was removed after confirming outlier status via Hampel and Grubbs outlier testing (both p<0.01).

### Clinical variables

Available deidentified clinical variables were collected from patients and from chart review during their time in the emergency department. Clinical variables were categorized to create combined variables such as the number of chronic conditions or the number of presenting symptoms. The full breakdown of clinical variable categorization can be found in *Figure 2—source data 1*.

### Data analysis

Data analysis and visualization were generated using R (*R Development Core Team, 2020*). The var-rank package (*Kratzer and Furrer, 2020*) was used to apply a minimum-redundancy maximum-relevance mutual information algorithm. The algorithm classifies the amount of information each cytokine and clinical variable can provide about the outcome variable, CSS. Each cytokine variable was discretized into two clusters – either high or low analyte concentration in pg/mL – using k-means clustering to minimize within-variable entropy and, thus, over-fitting. This algorithm partitions each data point into the cluster (high or low analyte concentration) with the nearest mean. Clinical variables and cytokine levels were used to predict CSS. The first variable was selected for local optimum relevance by a greedy algorithm. All subsequent variables were ordered to maximize relevancy and minimize redundancy. The ordering was robust to leave-one-out cross-validation. For each cytokine, one-way ANOVA with Tukey's honest significant difference test and Šidák correction for multiple comparisons was used to compare plasma cytokine levels among CSS groups.

## Acknowledgements

The work was supported by a Brown University COVID-19 Seed Grant (to WSE-D). The COVID-19 Biobank through which plasma samples were obtained was supported by Institutional Development Award Number U54GM115677 from the National Institute of General Medical Sciences of the National Institutes of Health, which funds Advance Clinical and Translational Research (Advance-CTR). The content is solely the responsibility of the authors and does not necessarily represent the official views of the National Institutes of Health. WSE-D is an American Cancer Society Research Professor.

## Additional information

### Competing interests

Wafik S El-Deiry: Senior Editor, *eLife*. The other authors declare that no competing interests exist.

### Funding

| Funder | Grant reference number | Author |
| --- | --- | --- |
| Brown University | | Wafik S El-Deiry |
| National Institute of General Medical Sciences | U54GM115677 | Kelsey E Huntington<br>Anna D Louie<br>Chun Geun Lee<br>Jack A Elias<br>Eric A Ross<br>Wafik S El-Deiry |

The funders had no role in study design, data collection and interpretation, or the decision to submit the work for publication.

## Author contributions
Kelsey E Huntington, Conceptualization, Data curation, Formal analysis, Investigation, Methodology, Writing - original draft, Writing - review and editing; Anna D Louie, Conceptualization, Formal analysis, Investigation, Visualization, Methodology, Writing - original draft, Writing - review and editing; Chun Geun Lee, Jack A Elias, Conceptualization, Resources, Investigation, Writing - review and editing; Eric A Ross, Formal analysis, Methodology, Writing - review and editing; Wafik S El-Deiry, Conceptualization, Resources, Supervision, Funding acquisition, Investigation, Methodology, Writing - original draft, Project administration, Writing - review and editing

## Author ORCIDs
Kelsey E Huntington (iD) https://orcid.org/0000-0002-5810-8220
Anna D Louie (iD) https://orcid.org/0000-0001-8394-6106
Wafik S El-Deiry (iD) https://orcid.org/0000-0002-9577-8266

## Decision letter and Author response
Decision letter https://doi.org/10.7554/eLife.64958.sa1
Author response https://doi.org/10.7554/eLife.64958.sa2

# Additional files

## Supplementary files
• Source code 1. Cytokine Mutual Information R Script.

• Transparent reporting form

## Data availability
Source data and source code files have been provided.

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
