## [Decision Letter]

**Acceptance summary:**

It's now known that SARS-CoV-2 infection can cause numerous changes in cytokines, and it has been hypothesized that in critically ill patients these changes are correlated with the severity of disease. This paper is of interest to the overall COVID-19 community in that presents an analysis of cytokines that identifies Macrophage Activation Syndrome as a correlate with disease severity.

**Decision letter after peer review:**

Thank you for submitting your article "Cytokine ranking via mutual information algorithm correlates cytokine profiles with disease severity in COVID-19" for consideration by *eLife*. Your article has been reviewed by two peer reviewers, one of whom is a member of our Board of Reviewing Editors, and the evaluation has been overseen by Mone Zaidi as the Senior Editor. The reviewers have opted to remain anonymous.

The reviewers have discussed the reviews with one another and the Reviewing Editor has drafted this decision to help you prepare a revised submission.

Summary:

The premise behind this manuscript is important and timely both for scientists focused on better understanding the variable immune response against the severe acute respiratory syndrome-coronavirus-2 (SARS-CoV-2) in different individuals and for physicians intent on improving the outcomes for their SARS-CoV-2 infected patients. The paper correlates the COVID-19 Severity Score (CSS) with a cytokine array along with demographic factors and a chest x-ray to see which components are predictive of severe COVID-19. The analysis is well presented and the experiments are straight forward in nature. Overall, the data are collected, controlled, and analyzed properly, and support the conclusions of the manuscript within the current context. If replicated in follow-on studies of independent patients, the findings may ultimately help to inform the development of improved diagnostics and therapeutics.

Essential revisions:

Overall the comments from the reviewers were positive, and suggest that the manuscript can be further strengthened by the revisions that are mostly explanatory in nature as indicated below.

1) Please include data on common lab parameters if available. For example, if there is data on CRP or fibrinogen, that would be interesting to add. Likewise, since the absolute monocyte count and the relative percentage of monocytes in the total white blood cell count are standard clinical assays obtained routinely in all SARS-CoV-2 infected patients presenting to hospital EDs for evaluating the possibility of progression in the less severely affected subset of subjects as well as in the subset of more severely affected subjects at presentation, can you discuss what you found when these two immunologically-relevant cellular parameters were included in the analysis you performed. If you did not analyze them, can you explain why not? Either way, a sentence or two should be included at this point in the Discussion section that these routinely measured cellular hematologic parameters would be a logical biomarker to explore in light of your main findings and conclusions, which include M-CSF as one of the four cytokines you identified that increase predictiveness.

2) Please state when the patient-derived blood samples were obtained. This is critical information and should be described here. Based on your use of "presenting" in the title and other places in the text, we are quite sure that the patient samples were collected on their arrival in the Emergency Dept of the Rhode Island Hospital, which would also have to be the case if the samples from all of the subjects (other than the normal subjects whose samples were purchased) were supposed to have been drawn at the same clinically-relevant point-in-time, i.e. as some of SARS-CoV-2 infected patients were asymptomatic and not admitted to the hospital, etc. Please also mention whether all of the patients were unrelated or if any were related, as this would then require accounting for the non-independence of the data generated from them, although this is mainly important for genetic association studies.

3) Please describe the centrifugation parameters employed including the g-force and length of time of the spin and mention what was the exact sample that was obtained. Was it, for example, citrated plasma samples used for coagulation studies? If so, why were the tubes needing to be spun again as whole blood samples centrifuged to remove aliquots of citrated plasma are spun at high speed two times to ensure platelet poor plasma which would also ensure cellular debris would have been removed.

4) The text of the manuscript, both in the Materials and methods section and elsewhere, creates the impression you performed a multiplex assay to measure the levels of the 15 cytokines listed in each study subject's plasma sample. But here, in the Materials and methods section, you mention (for the first time) that it is each subject's "production of" cytokines in the culture supernatant" which was used for the multiplex measurement. Please provide details as to: (i) what blood cell type or types (e.g., PBMCs) were used; (ii) how you isolated the blood cell(s); (iii) how you cultured them, i.e. in which medium and under what atmosphere they were cultured, and for how long, etc.; and (iv) how the conditioned medium was collected, processed and assayed.

5) Please explain briefly if and how in these statistical analyses you accounted for the issue of multiple testing (i.e., for the multiple cytokine measures).

6) It is important to include information on the well-known race- and ethnicity-based disparities in case fatality rates (CFRs) in African Americans and Mexican Americans. Please also include it as one of your "clinical variables" in your analysis. If you did incorporate it in your analysis, please add text to discuss it. If you did not, please discuss why not?

7) Figure 3 is cited in the text of the manuscript proper before Figure 2 is cited. Switch the numbering please.

8) In the legend for Figure 3B the labels for the y-axis ("years") and x-axis ("M-CSF concentration in pg/mL") are reversed to what is in the actual figure. Please correct this discrepancy.

9) It would be helpful to also show in Figure 3C the ROC curve(s) for predicting CSS 4-5 using age alone, and possibly, age together with only M-CSF levels.

10) Please correct the stylistic points and/or typos which we have highlighted yellow and/or struck through in the marked up word version of your manuscript.

---

## [Author Response]

Essential revisions:Overall the comments from the reviewers were positive, and suggest that the manuscript can be further strengthened by the minor revisions that are mostly explanatory in nature as indicated below.1) Please include data on common lab parameters if available. For example, if there is data on CRP or fibrinogen, that would be interesting to add. Likewise, since the absolute monocyte count and the relative percentage of monocytes in the total white blood cell count are standard clinical assays obtained routinely in all SARS-CoV-2 infected patients presenting to hospital EDs for evaluating the possibility of progression in the less severely affected subset of subjects as well as in the subset of more severely affected subjects at presentation, can you discuss what you found when these two immunologically-relevant cellular parameters were included in the analysis you performed. If you did not analyze them, can you explain why not? Either way, a sentence or two should be included at this point in the Discussion section that these routinely measured cellular hematologic parameters would be a logical biomarker to explore in light of your main findings and conclusions, which include M-CSF as one of the four cytokines you identified that increase predictiveness.

One of the biggest limitations of our data set is that the deidentified data of this biobank lacks certain parameters, such as common lab parameters like complete blood count, CRP or fibrinogen. While privacy issues limit our ability to obtain that data for this study, we have updated our Discussion to include this avenue for future research.

2) Please state when the patient-derived blood samples were obtained. This is critical information and should be described here. Based on your use of "presenting" in the title and other places in the text, we are quite sure that the patient samples were collected on their arrival in the Emergency Dept of the Rhode Island Hospital, which would also have to be the case if the samples from all of the subjects (other than the normal subjects whose samples were purchased) were supposed to have been drawn at the same clinically-relevant point-in-time, i.e. as some of SARS-CoV-2 infected patients were asymptomatic and not admitted to the hospital, etc. Please also mention whether all of the patients were unrelated or if any were related, as this would then require accounting for the non-independence of the data generated from them, although this is mainly important for genetic association studies.

In the Materials and methods section we have clarified when the patient-derived blood samples were obtained. As intuited by the reviewers, all samples (other than purchased normal samples) were indeed obtained on the patient’s arrival to the emergency department of Rhode Island Hospital. Unfortunately, due to the limited data present in the deidentified data set, we are unable to determine if any of the patients were related.

3) Please describe the centrifugation parameters employed including the g-force and length of time of the spin and mention what was the exact sample that was obtained. Was it, for example, citrated plasma samples used for coagulation studies? If so, why were the tubes needing to be spun again as whole blood samples centrifuged to remove aliquots of citrated plasma are spun at high speed two times to ensure platelet poor plasma which would also ensure cellular debris would have been removed.

The centrifugation parameters including g-force (14,000 rpm) and length of spin (10 minutes) have been added to the Materials and methods section. Although samples were platelet-poor plasma, and thus should already have been free of debris, the Luminex operating instructions suggest that all samples, regardless of their initial processing, be spun again immediately before being run to ensure they are free of debris and prevent potential clogs of the instrument.

4) The text of the manuscript, both in the Materials and methods section and elsewhere, creates the impression you performed a multiplex assay to measure the levels of the 15 cytokines listed in each study subject's plasma sample. But here, in the Materials and methods section, you mention (for the first time) that it is each subject's "production of" cytokines in the culture supernatant" which was used for the multiplex measurement. Please provide details as to: (i) what blood cell type or types (e.g., PBMCs) were used; (ii) how you isolated the blood cell(s); (iii) how you cultured them, i.e. in which medium and under what atmosphere they were cultured, and for how long, etc.; and (iv) how the conditioned medium was collected, processed and assayed.

Thank you for pointing out this inaccuracy in the Materials and methods section. The multiplex assay measured the levels of the 15 cytokines on each study subjects’ plasma sample. The Materials and methods have been updated to accurately reflect the experiment performed.

5) Please explain briefly if and how in these statistical analyses you accounted for the issue of multiple testing (i.e., for the multiple cytokine measures).

A Sidak correction was applied to correct for multiple testing for the multiple cytokine measures. The text and figures have been updated to reflect this.

6) It is important to include information on the well-known race- and ethnicity-based disparities in case fatality rates (CFRs) in African Americans and Mexican Americans. Please also include it as one of your "clinical variables" in your analysis. If you did incorporate it in your analysis, please add text to discuss it. If you did not, please discuss why not?

Race was included as a clinical variable in the mutual information algorithm. Unfortunately, the small sample size of our data set does not give it the power to identify race-based differences in COVID-19 severity. Indeed, the mutual information algorithm found the information provided by race was redundant after considering other factors such as age, symptoms, insurance status, number of abnormal vital signs, and chest x-ray abnormalities. We have updated our Discussion section to include discussion of race and COVID-19. In Author response image 1 is a race/ethnicity scatter plot if it is of interest to the reviewers.

7) Figure 3 is cited in the text of the manuscript proper before Figure 2 is cited. Switch the numbering please.

The numbering of these two figures has been corrected.

8) In the legend for Figure 3B the labels for the y-axis ("years") and x-axis ("M-CSF concentration in pg/mL") are reversed to what is in the actual figure. Please correct this discrepancy.

This discrepancy has been corrected.

9) It would be helpful to also show in Figure 3C the ROC curve(s) for predicting CSS 4-5 using age alone, and possibly, age together with only M-CSF levels.

The ROC curve for predicting CSS 4-5 found in Figure 2 has been updated to include a comparison curve predicting CSS 4-5 using age alone. This age ROC curve has an AUC of 0.74.

10) Please correct the stylistic points and/or typos which we have highlighted yellow and/or struck through in the marked up word version of your manuscript.

Stylistic points and typos have been addressed.